# Eating Behaviour Changes during the COVID-19 Pandemic: A Systematic Review of Longitudinal Studies

**DOI:** 10.3390/ijerph182111130

**Published:** 2021-10-22

**Authors:** Cristina González-Monroy, Irene Gómez-Gómez, Cristian M. Olarte-Sánchez, Emma Motrico

**Affiliations:** Department of Psychology, Universidad Loyola Andalucía, Dos Hermanas, 41704 Seville, Spain; cgonzalezmonrroy@al.uloyola.es (C.G.-M.); igomezg@uloyola.es (I.G.-G.); cmolarte@uloyola.es (C.M.O.-S.)

**Keywords:** eating behaviour, COVID-19, systematic review, longitudinal studies

## Abstract

Eating behaviour is a complex construct that is liable to be modified by external factors. Due to the outbreak of coronavirus disease 2019 (COVID-19), many restrictive measures were carried out with the aim of reducing the impact of this disease. As a result, lifestyles were disrupted, which could affect eating behaviours. The aim of this systematic review of longitudinal studies was to assess changes in eating behaviour during the COVID-19 pandemic by establishing a comparison of eating behaviours before and after the outbreak of the pandemic. This study followed the PRISMA guidelines (PROSPERO: CRD42020203246), whereas to assess the quality of the studies, the Newcastle-Ottawa Quality Assessment Scale (NOS) was applied. Out of a set of 826 studies, 23 were included in this systematic review. The main findings provided information about a shift towards modified eating behaviours, characterized by an increased snack frequency and a preference for sweets and ultra-processed food rather than fruits, vegetables, and fresh food. Additionally, an increased alcohol consumption was found among different countries. Consequently, adherence to healthy diets decreased. These findings are relevant to future policies and strategies to assess nutrition in cases of alarming situations such as the current COVID-19 pandemic.

## 1. Introduction

The appearance of coronavirus disease 2019 (COVID-19), an infectious disease caused by a coronavirus [1], has led to serious changes worldwide from late 2019 to the present day. Not only relationship patterns but also lifestyle habits have changed due to lockdown conditions, social distancing, reduced capacity in enclosed spaces and curfews proposed by governments. Although the aim of this measure was to fight the spread of the pandemic, the impact on daily life has been immense [2,3]. Since the first wave of the COVID-19 pandemic, many governments have ordered their population to stay at home while following security measures. As a result, people from many countries had to eat all of their meals at home while adapting their physical activity indoors due to not being allowed to go to public places or even to the gym [4,5]. In other cases, lifestyle habits suffered abrupt changes due to a lack of social services such as school cantines [6], and food insecurity [7]. As a result, eating behaviours could change across age groups [8,9,10].

Eating behaviours may be defined as food choices that are moderated by consumption trends, personal preferences, specific diets and calorie counting [11,12]. Regarding this definition, some cross-sectional studies have been carried out since the outbreak of the pandemic with the aim of achieving this objective [8,9,10,13,14,15,16,17]. However, as cross-sectional studies cannot identify differences in eating behaviour in comparison to previous baselines before the pandemic, bias can appear when trying to interpret the results [18,19,20,21,22]. This limitation appears to have been solved by summarizing the information in a few systematic reviews that have been carried out with the aim of exploring the impact of the pandemic on weight-related behaviours that include not only eating behaviours but also politics affecting the food supply and the health consequences in terms of nutritional status [18,19,20,21,22]. In this sense, a wide range of eating patterns that include both food restriction and loss of control over eating have been found, especially when analysing unhealthy food consumption [19,20]. This information remains relevant for people without comorbidities but also for vulnerable groups such as the older population and people with obesity that have experienced more acute changes in their eating behaviour [21,22].

Specifically, it has been found that the COVID-19 pandemic has had a global impact on daily diet among adults [19]. Although it has not been possible to establish a correlation between weight gain and changes on eating behaviour, an increased appetite accompanied by a higher consumption of snacks and a greater number of daily meals have been found. In the case of children, results appear to point at similar conclusions as the closure of school canteens has affected many families [6]. This situation has worsened the quality of diet in families with little resources as well as in families in which all members were workers. Finally, regarding dependent persons such as the elderly, people with morbid obesity and people with other type of disabilities, scientific evidence has linked preventive measures of physical contact restriction to changes in eating behaviour. Those changes have been associated to a significant decline in health nutrition [20].

This panorama has been worse in developing countries, where the outbreak of the COVID-19 pandemic has enhanced the need of proper define ‘food security’. Although apparently there was ‘availability’ and ‘accessibility’ to food supplies, many developing countries had no adequate supply chains. Consequently, families were not properly supplied with food, so they had to change their eating behaviours, which, in turn, contributed to a decline in health nutrition [7]. It is important to establish this dichotomy when talking about ‘eating behaviour changes during the COVID-19 pandemic’ because decision making about food may be motivated by different reasons, although the outbreak of the COVID-19 pandemic primarily triggered those changes.

In light of the above, the results suggest the need to properly understand the changes in eating behaviour that occurred during the development of the COVID-19 pandemic by focusing on longitudinal studies that can address changes by establishing comparisons before and after the outbreak of the pandemic. Therefore, the main objective of this study was to undertake a systematic review from September 2019 to July 2021 to evaluate eating behaviour changes, taking into account the following research question: which changes have occurred in eating behaviour during the COVID-19 pandemic? It is expected to produce an accurate description of those changes to better understand their consequences. These findings remain relevant to future policies and strategies to assess nutrition in cases of alarming situations such as the current COVID-19 pandemic.

## 2. Materials and Methods

This review followed the Preferred Reporting Items for Systematic Reviews and Meta-Analyses (PRISMA) guidelines [23]. Additionally, it was registered in the International Prospective Register of Systematic Reviews database (PROSPERO) (registration number CRD42020203246).

### 2.1. Search Strategy

The search strategy was defined by establishing the research question as well as the inclusion and exclusion criteria according to the PICO (Population/Participants, Intervention, Comparison and Outcomes) model [24], as shown in Table 1.

The search was first piloted in the PubMed database on 27 July 2021 (as shown in Table 2) and filtered since 31 December 2019. Then, it was replicated in three other databases: Cochrane Library, PsycInfo and Web of Science. Then, two clusters of search terms were designed. The first included terms related to COVID-19, while the second was related to eating behaviour changes. To develop a representative advanced search, the selected words were chosen from the Medical Subject Headings (MeSH) thesaurus. Additionally, lists of references from previous studies that were selected as representative of this review were examined.

### 2.2. Study Selection Process

This systematic review comprised 3 main steps. First, duplicated content was identified and deleted. Second, by following the inclusion/exclusion criteria, the studies were reviewed by title/abstract. Finally, those that passed the screening were reviewed by a full-text reading. All of these steps were conducted independently by two reviewers, and in cases of disagreement, a third reviewer was assigned to resolve any discrepancies. The inter-agreement of the total selection between reviewers was almost perfect [25] (Cohen Kappa Index = 0.85; 95% CI, 0.49 to 1.00).

## 3. Results

### 3.1. Study Selection

The search strategy led to a set of 816 studies. In addition, 10 studies were identified after examining the lists of references from the representative studies. Consequently, the first stage of the revision gathered 826 studies. Of those, 106 studies met the inclusion criteria under the title/abstract review. After a full-text reading, 83 studies were removed due to not meeting the inclusion criteria. Thus, 23 studies passed the screening process, which led to the next phase. The screening process is summarized in Figure 1.

### 3.2. Study Characteristics

The characteristics of the selected studies are summarized in Table 3. The variables are taken for in-depth analysis from the paragraphs below.

#### 3.2.1. Country and Design of the Studies

The twenty-three longitudinal studies included the following countries: India [26], Australia [27,28], Spain [29,30], Italy [31,32,33,34,35], Brazil [36], France [37], China [38,39,40,41], the United States [42], Canada [43], Poland [44], the United Kingdom [45,46] and Japan [47,48].

Twelve [26,27,28,29,30,36,37,42,43,45,46,48] (50%) were prospective studies, and eleven [31,32,33,34,35,38,39,40,41,44,47] (42.86%) were retrospective studies. Additionally, the length of follow-ups varied between 2 [32,35,47] and 54 weeks [28,42].

#### 3.2.2. Target Population

Although studies mainly focused on adult populations over 18 years old with no comorbidities [28,35,36,37,38,42,43,44,46,48], specific subgroups that might be taken into account were found: people diagnosed with diabetes mellitus [26,32,47], young people [27,29,30,31,34,39,40,41], people with obesity [33] and people in vulnerable situations [45]. The average of mean ages for twenty-three longitudinal studies was 24.19 years (SD = 15.3).

#### 3.2.3. Sample Size

Due to the differences among the studies, sample sizes varied between 34 [47] and 37,252 [37] participants (media *n* = 4918; IQR 112 to 10,082). However, the experimental mortality was not significant except for one study [27], which lost almost 44.84% of the participants. In addition, the ratio between males and females was relatively equal, with the exception of five studies in which samples were comprised mainly of females [27,33,36,41,43]. This systematic review evaluated a total of 116,952 participants.

#### 3.2.4. Assessment Tools

All the studies used different recording measures to assess target eating behaviour changes. In this sense, many of them were standardized validated scales, such as the NSW Centre for Public Health Nutrition [27], The Mediterranean Diet Quality Index for children and teenagers (KIDMED) [29], the 12-items lifestyle questionnaire [31], the House hold Dietary Diversity Score (HDDS) [38], the COVID-19 Impact on Lifestyle Change Survey (COINLICS) [39], the Diet Screener Questionnaire, the Dietary Questionnaire for Epidemiological Studies (DQUES and diet v3.2; Cancer Council Victoria) [28], the Food and Beverages Diet frequency questionnaire [40], the Modified Food frequency Questionnaire (FFQ-6) [44], and the Mediterranean diet (Med-Diet) questionnaire [30]; whereas the others used scales that were created for the specific purpose of their research [26,32,33,34,35,36,41,47]. Additionally, self-reports [37,43,45,46] were used in order to follow eating changes among participants. One study used ecological momentary assessment through the CALO mama health app for this purpose [48].

### 3.3. Objectives, Outcomes, and Results

Table 4 summarizes the objectives, outcomes, and results of the studies, as shown below.

#### 3.3.1. Objectives

All of the studies had a common objective, assessing eating behaviour changes caused by the COVID-19 pandemic. In this sense, sixteen studies focused on eating behaviour changes during lockdown conditions [26,29,31,32,33,34,35,37,38,39,40,41,42,43,45,46], while seven studies focused their research into the entire spectrum of the pandemic [27,28,30,36,44,47,48].

#### 3.3.2. Outcomes

Out of twenty-three studies, eighteen studies pursued eating behaviour as a primary or main outcome [27,28,32,33,34,35,36,37,38,39,40,42,43,44,45,46,47,48], while five studies included it as a secondary outcome [26,29,30,31,41].

#### 3.3.3. Results

All of the studies recorded eating behaviour changes during the COVID-19 pandemic in comparison to previous baselines before the pandemic, concluding that there were changes in the type of diet. Two studies found that there were changes in mealtimes [26,37], whereas three reported a more frequent food intake [26,35,39] and four an increased amount of food eaten [26,31,46,47]. In particular, six studies highlighted a greater consumption of snacks [33,34,35,41,44,47]. In this sense, five studies established that there was a preference for sweets [32,33,34,35,44] while two concluded that wheat products were more frequently consume after the COVID-19 pandemic [39,40]. In the case of beverages, two studies found a higher daily intake [35,40] while another one reported a decreased intake [39]. Specifically, for alcohol, six studies reported an increased consumption [28,38,41,44,45,48].

Otherwise, four studies reported a decreased consumption of fruit and vegetables [39,40,45,48], two that there were no substantial changes [27,31], and two that the daily intake got increased [30,36]. Regarding this, three studies found a decreased consumption of fresh food [37,39,40] and four other studies reported a higher intake of ultra-processed food [35,36,37,47]. Meat consumption got decreased according to two studies [40,42], but also got increased regarding three studies [30,31,48].

Finally, three studies resolved that negative changes in eating behaviour were found due to a lower adherence to healthy diets [29,35,46], meanwhile two reported adherence to healthy diet [30,43] and one study declared no changes in dietary habits just except for a reduction in the amount of meat eaten [42].

### 3.4. Quality of Studies

To assess the quality of the studies, the Newcastle-Ottawa Quality Assessment Scale (NOS) [49] was used. This tool was developed with the aim of being helpful for systematic reviews so that quality criteria can be reached while ensuring a low risk of bias. NOS uses three main categories: selection, comparability and outcome. Each category contains several items about the characteristics of observational studies, each of which includes several answers. Among those answers, at least one is accompanied by a star (☆). As stars symbolize a low risk of bias, once the scale is completed, it is necessary to sum them. If the result is less than 7 stars, there might be a high risk of bias. Additionally, in the case of reviewers selecting answers that do not include a star, it is important to check the possible biases of the study. Therefore, NOS scores might be categorized into three groups: very high risk of bias (0–3 stars), high risk of bias (4–6 stars) and low risk of bias (7–9 stars) [50].

The quality of each study was evaluated independently by two reviewers. A third reviewer was assigned in case of disagreement. The final consensus is shown in Table 5.

To quantify the degree of agreement between reviewers regarding overall quality assessment, the Cohen Kappa Index was calculated. The results showed that there was a good agreement level (Cohen Kappa Index = 0.74; 95% CI, 0.60 to 0.89) regarding the specific score of each study. By considering this, it was possible to conclude that there was a low risk of bias in all twenty-three studies.

## 4. Discussion

The current systematic review, which included twenty-three studies, provides a comprehensive overview of eating behaviour characteristics associated with the COVID-19 pandemic. Consistent with previous reviews, several outcomes, such as overeating [26,31,46,47] and the influence of personal preferences on food choices [32,33,34,35,44], have been observed. However, this systematic review also found that the frequency of meals is not associated with the amount of food eaten [32,36,40]. Additionally, variables, such as gender and age, that did not show a correlation with specific eating behaviours [29], need to be considered, as well as the influence of mental health [41,45,46]. In the case of work status, one study found that people who spend more time at home consumed a greater amount of self-made food [48] something that appears to be correlated with the availability of food delivery services [51], which are a faster option for people who is in a rush.

However, when referring to population with no comorbidities, results may generate controversy. Three studies reported patterns of stability on eating behaviour despite the outbreak of the pandemic [27,28,42], while two studies even reported improvements in the adherence to healthy diets [38,43]. In spite of that, results showed a more frequent intake of food, an increased consumption of ultra-processed food and a higher caloric intake due to a more frequent alcohol consumption [35,37,44,46,48].

Regarding specific subgroups, people with diabetes appeared to increase the daily amount of food eaten. In this sense, unhealthy food products such as sweets and starchy food were common among this population [26,32,47]. People with obesity showed similar trends by reporting a significant increase in the amount and frequency of unhealthy food products [33]. In the case of young people, a lower adherence to healthy diets such as the Mediterranean Diet [29] was found due to an increased intake of food, a preference for snacks and a lack of fruit and vegetables intake [27,29,30,31,34,39,40,41]. On the contrary, vulnerable population showed a reduced intake of food, contrasting with an increased alcohol consumption [45]. In all these cases, results did not show changes towards a healthier diet, what may emphasize the need to protect vulnerable population from risk situations that might affect health or, in this case, nutritional status.

By considering all this information, it is possible to confirm the existence of changes in eating behaviour during the COVID-19 pandemic. To the best of our knowledge, this is the first systematic review that examined these trends by considering longitudinal studies. With this design, it is possible to analyse the changes of eating behaviour in comparison with previous baselines before the pandemic, which made it possible to establish the level of significance of this phenomenon. Additionally, the screening process that was applied ensured the quality of this review, as shown in the NOS results of each study. Therefore, it might be possible to accept the relevance of this review on this topic.

Even so, several limitations should be considered when interpreting these results. First, due to the heterogeneity of the designs as well as the small set of papers used, it is important to consider this review as a first approach to eating behaviour changes during the COVID-19 pandemic. Its results may be useful to consider when undertaking future directions of reviews when more studies become available. Second, and according to the above, due to the novelty of this phenomenon, there were no comparators apart from within-subject comparability (prospective and retrospective), which is a limitation when trying to distinguish between the influence of the pandemic (lockdowns, social distance, isolation, uncertainty about the future, etc.) and any other external factor. Longitudinal between-subject studies may be necessary to dismiss possible bias caused by external factors. Third, this systematic review included both clinical and non-clinical population what may disturb results due to additional variables that has not been measured. Consequently, any interpretation should be made carefully, as the effect size might not represent society-wide eating behaviour, although previous studies have established common eating behaviour changes among clinical and non-clinical populations [21,22]. Finally, although one of the strengths of this review is that different countries have been included, it might be important to consider that restrictive measures have been different among countries so that the COVID-19 pandemic might have had a greater or lesser impact depending on the policies of each country [36].

Keeping all of this in mind, a deep understanding beyond the results is still possible. Therefore, this review is useful as a benchmark that contributes to the current body of knowledge about the impact of the COVID-19 pandemic on daily lifestyles. In addition to confirming the appearance of changes in eating behaviour since the outbreak of the COVID-19 pandemic, this is the first systematic review that has achieved an accurate description of those changes by considering different countries. The results of this study will provide a reference to guide future research directions among those interested not only in this topic but also in specific eating patterns as well as in the differences between ‘amount of food’ and ‘snack frequency’. All of these approaches will lead to a better understanding of eating behaviour during the COVID-19 pandemic as well as contribute to future guidelines about health promotion.

### Implications for Government Policies in Nutrition

Due to the importance of eating behaviour as criteria on health and safety, it is important for governments to ensure healthy eating patterns among population through health education programmes and suitable access to food supply [52]. In this sense, nutrition-sensitive policies might seek participation from different social and economic sectors with the aim of reaching a proper change in eating behaviour [53]. Some examples of these policies would include: (a) access to education, (b) the promotion of healthy local food environments as well as social protection programmes, (c) the regulation of working conditions, (d) the improvement of menus at school cantines and (e) the regulation of advertising through a gender-based perspective that rejects beauty standards and encourages a healthy lifestyle [54].

This initiative would address the Sustainable Development Goals (SDGs), in particular, the SDG 2 which aims to ‘End hunger, achieve food security and improved nutrition and promote sustainable agriculture’ [55]. Nutrition plays an important role within international cooperation as hunger elimination together with the development of sustainable food systems might improve several environmental aspects of strategic relevance and interest [56]. This systematic review provides important information about food choices in case of worldwide alarming situations such as the COVID-19 pandemic that might be responsible for a lack of food supply and negative emotions [4,9]. By considering that, future programmes aimed at mindful eating and healthy habits might include guidelines towards resources of interest during state alarms, examples of balanced diets and their budgets, area-based lists of soup kitchens, and allowances. Not only that but also, educational programmes might be extended beyond health and politic fields by applying multi-strategy campaigns concerned with sustained public awareness on healthy eating [53]. This kind of approach might be suitable for developed and developing countries as its objective would be ensuring a proper use of resources as well as healthier choices. Evidence-based interventions have demonstrated the efficacy of nutritional education programmes into adaptative eating behaviour [57], what might involve an urgent call to action for governments and social sectors focused on health promotion. Some implications of this initiative would include a lower demand of healthcare resources [58], a greater productivity in the workplace [59], and a better quality of life among countries due to the access to basic needs [60].

## 5. Conclusions

The COVID-19 outbreak led to changes in eating behaviour, which may have become less healthy during the pandemic. Although these changes could be a result of uncertainty and discomfort, adverse effects on health, especially for vulnerable population, would emphasize the need to promote healthy habits through preventive interventions and social actions supported by governments. Additionally, with the aim of assessing a more accurate framework of the stages through which eating behaviour changes evolved during this period, further research should be carried out. In this context, it would be important to focus on food intake but also on alcohol consumption and its consequences. By this, not only may a progression of ‘eating behaviour during a state of alarm’ be developed but also a benchmark for future directions can be established that will help improve guidelines for achieving proper nutrition aimed at the new normality.

## Figures and Tables

**Figure 1 ijerph-18-11130-f001:**
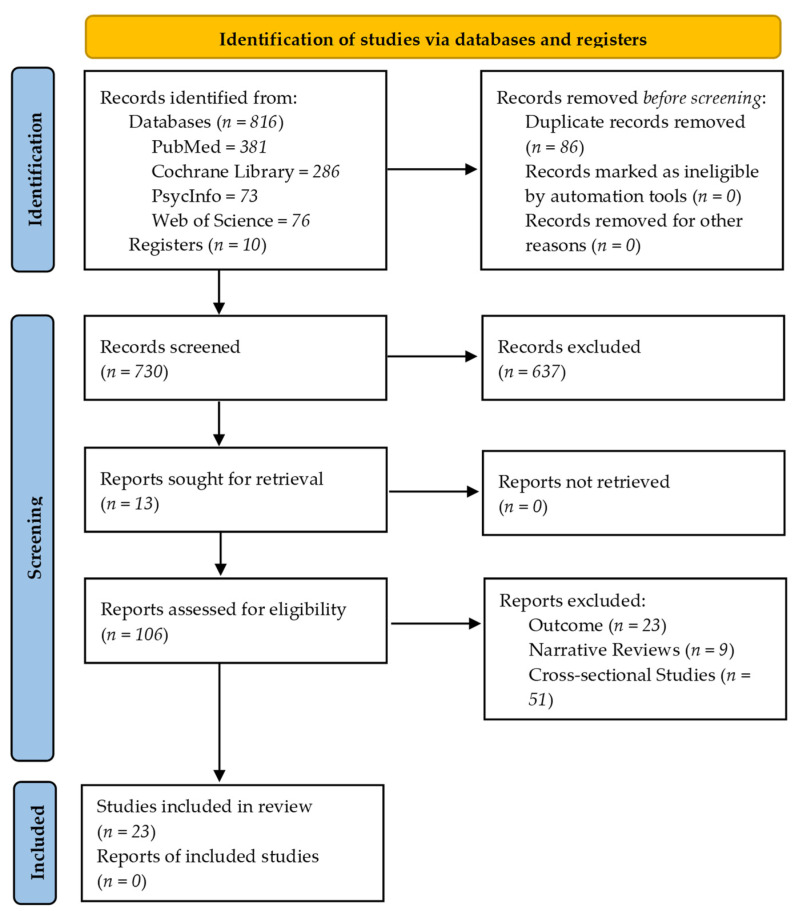
PRISMA flow diagram of the included and excluded studies throughout the systematic review protocol.

**Table 1 ijerph-18-11130-t001:** Inclusion and Exclusion Criteria.

	Inclusion Criteria	Exclusion Criteria
Participants	All participants	None
Outcome	Eating behaviour changes during the COVID-19 pandemic.	Those studies which do not assess pre/post-records of data among participants.
Type of study	Longitudinal studies (prospective and retrospective)	Cross-sectional studies, reviews, letters to the editor, recommendations.
Language	All languages	None
Setting	All settings	None

**Table 2 ijerph-18-11130-t002:** Search Strategy in PubMed.

Search Strategy
((((((“COVID-19”[MeSH Terms]) OR “coronavirus”[MeSH Terms]) OR “quarantine”[MeSH Terms]) OR “coronavirus pandemic”[Title/Abstract]”) OR 2019-nCov”[Title/Abstract]) OR “confinement”[Title/Abstract]) OR “lockdown”[Title/Abstract]) AND (((((((((“feeding behavior”[MeSH Terms]) OR “diet, healthy”[MeSH Terms]) OR “diet*” [MeSH Terms]) “Diet, Food, and Nutrition”[MeSH Terms]) OR “healthy eating”[Title/Abstract]) OR “diet*”[Title/Abstract]) OR “dietary pattern”[Title/Abstract]) OR “food intake”[Title/Abstract]) OR “nutrit*”[Title/Abstract]) OR “diet change”[Title/Abstract] AND ((cohort [Title/Abstract]) OR (longitudinal [Title/Abstract]) OR (follow-up [Title/Abstract]) OR (prospective [Title/Abstract]) OR (retrospective [Title/Abstract]) OR (observational [Title/Abstract]))

**Table 3 ijerph-18-11130-t003:** Characteristics of the included studies.

First Author/Year/ Region (Country)	Design (Study Type)	Length of Follow-Up (Weeks)	Target Population (Age)	Inclusion Criteria	Exclusion Criteria	Sample Size at Baseline (Total *n*/Female (%))	Assessment Tool	Target Lifestyles	Sample Size at Follow-Up (Total *n*/Female (%))	Statistical Technique/s
Khare, (2020) Bhopal (India)	PS	3 weeks	Patients aged 18–65 years old with a diagnose of type 2 diabetes mellitus	Adult patients with previously diagnosed type 2 diabetes mellitus on treatment, and willing to participate in the study	Sick patient, patients with established microvascular or macrovascular complication of diabetes, pregnancy, and not willing to participate in the study	*n* = 143 Female = 36.36% (*n* = 52)	Questionnaire designed for this study	Diet, exercise, sleep, glycaemic control	*n* = 143 Female = 36.36% (*n* = 52)	Descriptive analysis (paired *t*-test)
Munasinghe, (2020) New South Wales (Australia)	PS	22 weeks	Adolescents aged 13–19 years of a Sydney population catchment	(1) Age 13–19 years old (2) living in Sidney	No	*n* = 1298 Female = 80.5% (*n* = 1045)	Self-reported dietary behaviour was measured using questions validated for adolescents by the NSW Centre for Public Health Nutrition	Physical activity, sedentary behaviour, dietary behaviour, screen time	*n* = 582 Female = 79.89% (*n* = 465)	Multivariate multilevel mixed-effect logistic regression models
Medrano, (2020) All regions (Spain)	PS	6 weeks	Spanish children aged 10–14 years old	Children and adolescents who were enrolled in a participating school were eligible for inclusion	Pupils with an intellectual or a physical disability that prevents response to the lifestyle questionnaires were initially excluded from the MUGI project	*n* = 291 Female = 47.8% (*n* = 139)	The Mediterranean Diet Quality Index for children and teenagers (KIDMED)	Physical activity, sleep, screen time and diet	*n* = 113 Female = 48.7% (*n* = 55)	(1) Dependent *t*-test (continuous variables) and (2) analysis of covariances (ANCOVA)
Pietrobelli, (2020) Verona (Italy)	RS	11 weeks	Children and adolescents aged 6–18 years old	Children and adolescents with overweight and obesity	No	*n* = 41 Female = 46.34% (*n* = 19)	12 items lifestyle questionnaire (sports activity participation, screen time, sleep behaviour, and eating habits, focusing on servings of red meat, pasta, snacks, fruits, and vegetables)	Physical activity, screen time, sleep behaviour and eating habits	*n* = 41 Female = 46.34% (*n* = 19)	(1) Descriptive statistics of the participant’s baseline, (2) paired *t*-tests, (3) correlation analysis, (4) independent two-sample *t*-tests
Caruso, (2020) Bari (Italy)	RS	2 weeks before and 2 weeks after lockdown	People ≥18 years old with a diagnose of type 1 diabetes mellitus	Both males and females older than 18 years old diagnosed with diabetes mellitus type 1 for at least 1 year. Being on an intensive insulin regimen for at least 6 months and Having their glucose data uploaded to the Libreview platform	History of acute myocardial infarction, stroke and percutaneous or surgical revascularization within the last 30 days. (2) Active neoplasia within the last 6 months and (3) Any other concomitant medical or psychological condition	*n* = 48 Female = 47.9% (*n* = 23)	Changes in dietary habits were assessed on a semi-quantitative basis (higher/lower/same frequency) with an ad hoc questionnaire	Physical activity and diet	*n* = 48 Female = 47.9% (*n* = 23)	(1) Two-tailed paired Student’s *t*-test (2) Wilcoxon matched-pairs signed-rank test, and (3) chi-square test
Pellegrini, (2020) Nothern Italy (Italy)	RS	4 weeks	Patients aged 18–75 years old	Patients of the Obesity Unit of the Diabetes and Metabolic Diseases Clinic of the Città dela Salute e della Scienza Hospital of Torino	Dropouts	*n* = 150 Female = 77.33% (*n* = 116)	A 12-question multiple-choice questionnaire	Physical activity and diet	*n* = 150 Female = 77.33% (*n* = 116)	(1) *T*-test for paired samples, (2) Mann-Whitney test (2 groups), (3) Kruskal-Wallis test (3 groups), (4) multiple regression model, and (6) regression path model
Martínez-Steele, (2020) All regions (Brazil)	PS	15 weeks	People ≥18 years old	People older than 18 years old residing in Brazil	No	*n* = 10,116 Female = 78% (*n* = 7895)	A simplified questionnaire about diet	Diet	*n* = 10,116 Female = 78% (*n* = 7895)	(1) Chi square, and (2) *t*-test
Deschasaux-Tanguy. (2020) All regions (France)	PS	24 weeks	French people ≥18 years old	Adults aged ≥18 years old living in France	No	*n* = 37,252 Female = 52.3% (*n* = 19.483)	Web-based 24-h dietary records	Food-related practices, physical activity, and body weight	*n* = 37,252 Female = 52.3% (*n* = 19.483)	(1) The Alternative Healthy Eating Index (AHEI)-2010 score, (2) NOVA classification, (3) multiple correspondence analysis, (4) ANCOVA models for variations in continuous variables, and (5) multivariable logistic regression models
Zhang (2020) All regions (China)	RS	20 weeks	Adults living in China aged 18–80 years old	Adults 18–80 years old living in mainland China	No	*n* = 1994 Female = 62.79% (*n* = 1252)	(1) A standardized survey, (2) the Household Dietary Diversity Score (HDDS), and (3) a series of questions regarding changes in dietary behaviours	Diet	*n* = 1994 Female = 62.79% (*n* = 1252)	(1) Univariate analysis, (2) A linear regression model, (3) K-means clustering analysis, and (4) Multivariable analysis
Yu. (2020) All regions (China)	RS	4 weeks	People from China aged 17–22 years old	Youth people under three educational attachments (i.e., high school, college or graduate) in China	No	*n* = 10,082 Female = 72.39% (*n* = 7298)	COVID-19 Impact on Lifestyle Change Survey (COINLICS)	Diet	*n* = 10,082 Female = 72.39% (*n* = 7298)	(1) Descriptive statistics to summarise characteristics of participants, (2) Pair *t*-test to compare the differences in frequency of food and beverage consumption, and (3) A factor analysis with orthogonal rotation to derive main dietary patterns before-after the pandemic
Barone, (2021) Pennsylvania (USA)	PS	54 weeks	Desk workers ≥18 years old	Participants in the RESET BP trial	No	*n* = 112 Female = 69% (*n* = 77)	Diet Screener Questionnaire	Sedentary behaviour, physical activity, sleep quality, and diet	*n* = 112 Female = 69% (*n* = 77)	(1) Paired samples, (2) McNemar tests, and (3) Bowker’s tests
Curtis, (2021) Adelaide (Australia)	PS	54 weeks	Healthy adults aged 18–65 years old	Adults residing in greater metropolitan Adelaide, having access to a Bluetooth-enabled mobile device or computer and home internet, proficiency in English, and ambulant	Pregnancy, having an implanted electronical medical device, or experiencing or receiving treatment for any life-threatening condition impacting daily lifestyle and health	*n* = 61 Female = 65.6% (*n* = 40)	Dietary Questionnaire for Epidemiological Studies (DQUES and diet v3.2; Cancer Council Victoria)	Physical activity, and diet	*n* = 61 Female = 65.6% (*n* = 40)	Mixed effects models
Jia, (2021) All regions (China)	RS	20 weeks	Chinese youth aged 15–28 years old	No	No	*n* = 10,082 female = 71.7% (*n* = 7320)	(1) Food frequency questionnaire, and (2) Beverages Diet frequency questionnaire	Diet	*n* = 10,082 Female = 71.7% (*n* = 7320)	(1) Chi square, and (2) *T*-test
Lamarche, (2021) Quebec (Canada)	PS	20 weeks	Adults living in Quebec ≥18 years old	Adults aged ≥18 years old with a residential address in the province of Quebec	Not being able to read and understand French or English. Not having access to the Internet (with a phone, tablet or computer), and not having an active email address. Living outside of the province of Quebec	*n* = 853 female = 87.2% (*n* = 744)	Self-administered web-based 24-h time dietary recall	Smoking, screen time, physical activity, and diet	*n* = 853 Female = 87.2% (*n* = 744)	Linear mixed regression models
Czenczek-Lewandowska. (2021) South-east Poland (Poland)	RS	5 weeks	Polish people in early adulthood aged 18–34 years old	People aged from 18 to 34 years of Polish nationality resident in south-east Poland throughout the pandemic period	No	*n* = 506 Female = 70.2% (*n* = 355)	The modified Food Frequency Questionnaire (FFQ-6)	Diet, physical activity and sedentary behaviour, and sleep quality	*n* = 506 Female = 70.2% (*n* = 355)	
Naughton. (2021) High deprivation areas of all regions (United Kigndom)	PS	12 weeks	Adult-vulnerable people from the United Kingdom ≥18 years old	Adults living in the United Kingdom. Those with a physical high-risk health condition for COVID-19 (in line with the UK National Health Service definitions), those living in a high deprivation area, and those with a self-reported mental health issue.	People without access to a smartphone	*n* = 1044 Female = 72.7% (*n* = 747) * Other = 0.2% (*n* = 2)	(1) An online survey, and (2) Daily health behaviour monitoring	Diet, physical activity, alcohol, smoking and vaping, and substance use behaviours	*n* = 1044 Female = 72.7% (*n* = 747) * Other = 0.2% (*n* = 2)	(1) Descriptive analyses for all health behaviour measures, (2) Multivariable regression models, and (3) Covariates
Imaz-Aramburu. (2021) Basque Country (Spain)	PS	48 weeks	Students from Spain ≥19 years old	Nursing and medical students at the beginning of the second year of their degree and a physiotherapy student at the beginning of the third year at the Faculty of Medicine and Nursing, University of the Basque Country, Spain	No	*n* = 267 Female = 76% (*n* = 203)	(1) An ad hoc self-administered questionnaire, (2) the Mediterranean diet (MedDiet) questionnaire	Diet, and physical activity	*n* = 267 Female = 76% (*n* = 203)	(1) Kolmogorov–Smirnov test, (2) a Wilcoxon signed-rank test to analyze the difference between MedDiet scores, moderate and intense physical activity, and eating habits, (3) Paired *t*-tests, and (4) a McNemar test to compare proportions between adherence to the Mediterranean Diet and physical activity frequency
Segre. (2021) Milan (Italy)	RS	3 weeks	Italian children and adolescents aged 6–14 years old	Primary or middle school students living in Milan	Not being able to attend Zoom Meetings	*n* = 82 Female = 46.3% (*n* = 38)	10–20 min-structured interviews	Distance learning, eating, and sleeping habits	*n* = 82 Female = 46.3% (*n* = 38)	Chi-square and Fisher’s exact test.
Maffoni. (2021) All regions (Italy)	RS	2 weeks	Italian people ≥18 years old	Adults >18 years residing in Italy	No	*n* = 1304 Female = 74.62% (*n* = 973)	(1) A 38 multiple-choice web-form survey in Google Forms, and (2) 10 multiple-choice items	Water intake, alcohol consumption, physical activity, diet	*n* = 1304 Female = 74.62% (*n* = 973)	(1) *T*-test, and (2) Multiple Regression Analyses
Hosomi. (2021) Kyoto (Japan)	RS	2 weeks	Patients diagnosed with Type 1 Diabetes Mellitus, aged 16–75 years old	Patients with Type 1 Diabetes Mellitus who visited the Department of Endocrinology and Metabolism of the Kyoto Prefectural University of Medicine	Patients with missing data	*n* = 34 Female = 67.65% (*n* = 23)	A self-administered questionnaire related to stress and lifestyle factors	Stress levels, sleep time, exercise intensity levels, total diet, snack, and prepared food intake.	*n* = 34 Female = 67.65% (*n* = 23)	(1) Descriptive analyses, (2) Spearman’s correlation, (3) and Pair *t*-test to evaluate differences before and after the pandemic
Herle. (2021) All regions (United Kingdom)	PS	5 weeks	Adults ≥18 years old	Adults living in the UK during the COVID-19 pandemic	No	*n* = 22,374 Female = 76% (*n* = 16,984)	Self-reported eating changes	Eating behaviour, mental health, physical health and body weight	*n* = 22,374 Female = 76% (*n* = 16,984)	(1) Descriptive statistics, and (2) Latent class growth analysis to derive trajectories of change in eating across lockdown
Dun. (2021) Zhejiang and Hunan (China)	RS	3 weeks	Chinese students aged 17–27 years old	Students from Hunan Traditional Chinese Medical College (Hunan, China) and from the Medical College of Jinhua Polytechnic (Zhejiang, China) who were free of chronic diseases and had measured body weight at the universities before the lockdown	No	*n* = 12,889 Female = 80.22% (*n* = 10,340)	An 81-item-online follow-up questionnaire	Perceptions of COVID-19, physical activity, sedentary time, meal frequency, alcohol consumption, and mental health	*n* = 12,889 Female = 80.22% (*n* = 10,340)	(1) Paired *t*-test and Wilcoxon signed-rank test, (2) A baseline body weight-adjusted linear regression for associations with dietary habits, and (3) A multivariate linear regression
Sato. (2021) All regions (Japan)	PS	19 weeks	Adults ≥18 years old	Adults living in Japan	No	*n* = 5929 Female = 69% (*n* = 4087)	CALO mama health app	Diet and physical activity	*n* = 5929 Female = 69% (*n* = 4087)	(1) Generalized linear mixed models, (2) A Poisson distribution was to the count of intake of the food groups, and (3) A negative binomial distribution was to the count of drinking alcohol

Note. Design: PS: Prospective Study; RS: Retrospective Study. Sample Size at Baseline (Total n/Female (%)): * Other: Non-binary/ Transgender participants.

**Table 4 ijerph-18-11130-t004:** Objective, outcomes and results of the included studies.

Authors	Objective	Outcome	Results
Khare et al. (2020)	To study the effect of lockdown on glycaemic control in diabetic patients and possible factors responsible for this	Dietary changes (secondary outcome)	Dietary changes were observed as a change in the type of diet, change in timing of meals, change in frequency of meals and change in the amount of diet.
Munasinghe et al. (2020)	To investigate changes in physical activity, dietary behaviours, and well-being during the early period of physical distancing policies	Dietary behaviour (primary outcome)	There were declines in fast food consumption following implementation of physical distancing but no substantial changes in fruit and vegetable consumption.
Medrano et al. (2020)	To examine the effects of the COVID-19 confinement on lifestyle behaviours and to assess the influence of social vulnerabilities on changes in lifestyle behaviours	Adherence to the Mediterranean Diet (primary outcome)	Children worsened their lifestyle behaviours during the COVID-19 confinement, specifically their adherence to the Mediterranean Diet. No significant differences were found behaviours between primary and secondary students.
Pietrobelli et al. (2020)	To test the hypothesis that factors contributing to weight gain among children and adolescents with overweight and obesity are exacerbated during a pandemic associated lockdown	(1) Number of meals eaten per day, (2) changes in vegetable and fruit intake (3) potato chips intake, (4) red meat intake and (5) sugary drinks (primary outcomes)	The number of meals eaten per day got increased. There were no changes in vegetable intake, and fruit intake increased. There was an increased consumptions of potato chips, red meat, and sugary drinks during the lockdown.
Caruso et al. (2020)	To assess lockdown-related changes in glucose control and variability and their association with psychological distress and lifestyle changes in type 1 diabetes mellitus patients	(1) Meals per day, (2) starchy foods, (3) sweets, (4) whole grains and (5) vegetables (secondary outcomes)	Patients increased the frequency of starchy foods and sweets consumption, respectively.
Pellegrini et al. (2020)	To evaluate the changes in weight and dietary habits in a sample of individuals with obesity after 1 month of enforced lockdown	Changes in dietary habits (primary outcome)	There was an increased consumption of snacks, cereals, and sweets.
Martínez-Steele et al. (2020)	To describe the dietary characteristic of a cohort immediately before and during the pandemic	Changes in dietary characteristics (primary outcome)	The consumption of vegetables, fruits and legumes increased on daily diet. There was a pattern of stability in the consumption of ultra-processed food, although the number of people who consumed it increased.
Deschasaux-Tanguy. (2020)	To explore potential changes in dietary intake, physical activity, body weight and food supply during the COVID-19 lockdown and how these differed according to individual characteristics	Determinants of nutrition-related behaviour (primary outcome)	Diet-related practices during COVID-19 lockdown were modified. Trouble to keep a regular mealtime schedule, more frequent snaking (at least 3 times a day, every day) and a decreased consumption of fresh food was reported. Ultra-processed food consumption increased.
Zhang. (2020)	To explore post-lockdown dietary behaviours and their effects on dietary diversity.	Dietary behaviour (primary outcome)	An increased consumption of seafood and dietary supplements was found. Also, frozen food and raw food consumption decreased, whereas a higher alcohol and vinegar intake was registered.
Yu. (2020)	To assess changes in dietary patterns among youths in China after COVID-19 lockdown	Dietary patterns (primary outcome)	Significant changes in the weekly frequency of major food intake after lockdown was reported. There was a decrease in rice intake while an increase consumption of other staple food such as fish, eggs, fresh vegetables, preserved vegetables, fresh fruit and dairy products was recorded. There was an increased consumption of wheat products and a decrease in the frequency of beverages intake.
Barone. (2021)	To study the longitudinal impact of COVID-19 on work practices, lifestyle and well-being among desk workers during COVID-19 lockdown	Dietary habits (primary outcome)	No changes in dietary habits were found, except for a reduced frequency of red meat consumption.
Curtis. (2021)	To examine changes in activity patterns, recreational physical activities, diet, weight and wellbeing from before to during COVID-19 restrictions	Dietary intake (primary outcome)	Total energy intake did not change, but a slightly lower percentage of energy from protein and a greater percentage of energy from alcohol were recorded.
Jia. (2021)	To measure changes in diet patterns during the COVID-19 lockdown among youths in China	Diet patterns (primary outcome)	Significant changes in diet patterns associated with more frequent consumption of wheat products, other staple foods, preserved vegetables, and tea and less frequent consumption of rice, meat, poultry, fresh vegetables and fruit, soybean and dairy products and sugar-sweetened beverages were found.
Lamarche. (2021)	To document the change in diet quality and in food insecurity observed during the COVID-19–related early lockdown	Diet quality (primary outcome)	There were small but significant increases in the following components: whole grains, greens and beans, refined grains (reduced consumption), total vegetables, total dairy, seafood and plant proteins, added sugar (reduced consumption), and total proteins. The overall diet quality slightly improved.
Czenczek-Lewandowska. (2021)	To assess whether and to what extent the outbreak of the COVID-19 pandemic influenced the health-promoting behaviours of young adults in terms of eating habits, physical activity, sedentary behaviours and sleep.	Eating habits (primary outcome)	Average consumption of sweets, snacks, cereal products increased. A significant higher intake of alcohol and fats was recorded.
Naughton. (2021)	To provide baseline cohort descriptives and assess change in health behaviours since the UK COVID-19 lockdown	Dietary behaviour (primary outcome)	There were reductions in the mean reported number of daily portions of vegetables and fruit but no change in reported portions of high sugar food consumed. In the case of alcohol consumption, there was an increase in the average intake per month.
Imaz-Aramburu. (2021)	To understand the influence that the pandemic exerted on the lifestyles of university students, in particular those enrolled in health sciences programs	Eating habits (secondary outcome)	There was a greater adherence to the Mediterranean Diet. The consumption of vegetables increased significantly during the pandemic whereas the consumption of fatty meats tended to increase. An increased consumption of nuts was recorded.
Segre. (2021)	To analyse the impact of the quarantine on students’ life in Milan, one of the Italian cities most affected by COVID-19	Eating habits (primary outcome)	The amount of food eaten increased, with a higher consumption of junk food, snacks and sweets.
Maffoni. (2021)	To investigate lifestyle habits and eating behaviours modifications in a sample of Italian adults during “Phase 1” COVID-19 pandemic home confinement	Eating behaviours (primary outcome)	Negative changes in eating behaviour were found. These included an increased consumption of desserts or sweets at lunch. Craving or eating between meals was reported as well as the higher consumption of unhealthy snacks, beverages and ultra-processed food.
Hosomi. (2021)	To investigate the effects of the COVID-19 pandemic on the glycaemic control and changes in stress levels and lifestyle in patients with T1D.	Diet (primary outcome)	Average amount of food intake got increased. The consumption of snacks and prepared food were also considerably increased.
Herle. (2021)	To describe how people’s eating behaviour (eating more, eating less, eating the same) changed over 8 weeks of lockdown in the UK	Eating behaviour (primary outcome)	Many individuals did experience changes to their food intake as well as lower adherence to healthy eating patterns
Dun. (2021)	To observe weight change in Chinese youth during a 4-month lockdown and the associations between weight change and COVID-19-related stress, anxiety, depression, physical activity, sedentary time and dietary habits.	Dietary habits (secondary outcome)	A higher alcohol consumption was found in particular among men. Daily snack frequency also increased.
Sato. (2021)	To study dietary changes during the pandemic while considering changes in work and life patterns	Dietary changes (primary outcome)	Working people tended to consume less fruits and dairy products but more meats and alcohol compared with non-workers. In particular, irregular workers consumed less vegetables, mushrooms and fish. The average daily number of self-made meals increased among all participants.

**Table 5 ijerph-18-11130-t005:** Quality appraisal studies.

Study ID	Selection				Comparability	Outcome			Total	Score
	Representativeness of the Exposed Cohort	Selection of the Non-Exposed Cohort	Ascertainment of Exposure	Demonstration That the Outcome of Interest Was Not Present at the Start of the Study	Comparability of Cohorts on the Basis of the Design or Analysis	Assessment of Outcome	Was Follow-Up Long Enough for Outcomes to Occur	Adequacy of the Follow-Up		
Khare et al. (2020)	☆	☆	☆	☆	☆☆	☆	☆	☆	9	Low risk of bias
Munasinghe et al. (2020)	☆	☆	☆		☆	☆	☆	☆	7	Low risk of bias
Caruso et al. (2020)	☆	☆	☆	☆	☆☆	☆ ☆		☆	9	Low risk of bias
Medrano et al. (2020)	☆	☆	☆	☆	☆	☆ ☆	☆	☆	9	Low risk of bias
Pietrobelli et al. (2020)	☆	☆		☆	☆	☆	☆	☆	7	Low risk of bias
Pellegrini et al. (2020)	☆	☆	☆	☆	☆	☆	☆	☆	8	Low risk of bias
Martínez-Steele et al. (2020)	☆	☆	☆	☆	☆	☆	☆	☆	8	Low risk of bias
Deschasaux-Tanguy et al. (2020)	☆	☆		☆	☆	☆	☆	☆	7	Low risk of bias
Zhang et al. (2020)	☆	☆	☆	☆	☆	☆ ☆	☆	☆	9	Low risk of bias
Yu et al. (2020)	☆	☆	☆	☆	☆	☆	☆		7	Low risk of bias
Barone et al. (2021)	☆	☆	☆	☆	☆	☆ ☆	☆		8	Low risk of bias
Curtis et al. (2021)	☆	☆		☆	☆	☆ ☆	☆	☆	8	Low risk of bias
Jia et al. (2021)	☆	☆			☆	☆ ☆	☆	☆	7	Low risk of bias
Lamarche et al. (2021)	☆	☆	☆	☆	☆	☆ ☆	☆	☆	9	Low risk of bias
Czenczek-Lewandowska et al. (2021)	☆	☆	☆	☆	☆	☆	☆		7	Low risk of bias
Naughton et al. (2021)	☆	☆	☆	☆	☆	☆ ☆	☆	☆	9	Low risk of bias
Imaz-Aramburu et al. (2021)	☆		☆	☆	☆	☆ ☆	☆	☆	8	Low risk of bias
Segre et al. (2021)	☆	☆	☆	☆	☆	☆	☆		7	Low risk of bias
Maffoni et al. (2021)	☆	☆		☆	☆	☆	☆	☆	7	Low risk of bias
Hosomi et al. (2021)	☆	☆	☆	☆	☆	☆	☆		7	Low risk of bias
Herle et al. (2021)	☆	☆		☆	☆	☆	☆	☆	7	Low risk of bias
Dun et al. (2021)	☆	☆	☆	☆	☆	☆		☆	7	Low risk of bias
Sato et al. (2021)	☆	☆		☆	☆	☆ ☆	☆	☆	8	Low risk of bias

Note. The following items could gather up to 2 stars (☆ ☆): ‘Representativeness of the exposed cohort’; ‘Ascertainment of exposure’; ‘Comparability of cohorts on the basis of the design or analysis’; ‘Assessment of outcome’; ‘Adequacy of follow-up’.

## Data Availability

Not aplicable.

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
