# Peer review of "Eating Behaviour Changes during the COVID-19 Pandemic: A Systematic Review of Longitudinal Studies"

_ijerph, 2021, doi:10.3390/ijerph182111130_

Round 1
Reviewer 1 Report
The authors brought up an important issue that related to covid-19 and found 7 studies within 6 months timeframe. From your results, those 7 studies are in different countries studying on people with different age, ethnicity, and healthy status, different government policies, and different stage of COVID spreading. Thus, I would suggest the author expand their search to the end of pandemic.
Reviewer 2 Report
Comments to the Author
The authors conducted a systematic review of cohort studies to compare changes in eating behavior during the COVID-19 pandemic before and after the pandemic.
The authors' attempt to summarize the changes in eating behavior due to the pandemic, with an emphasis on the pre-pandemic baseline, is commendable.
However, several issues need to be addressed with this review.
(1) How do the authors define a cohort study? It is desirable to describe it as a longitudinal study, since it is the only study that followed a cohort.
(2) The seven selected papers have different backgrounds in terms of age groups, underlying diseases, cultures, and time periods. I do not feel that a systematic review of seven studies with such different backgrounds and subjects is meaningful. I think it would be more meaningful to include cross-sectional studies and look at each background and subject in detail by stratum.
(3) It is stated that there is no change in the number of meals eaten as a new finding from the longitudinal study, but this cannot be said to be a finding that came about because of the longitudinal study.
(4) The exposure is described as a COVID-19 pandemic, but isn't it precisely a lockdown?
Reviewer 3 Report
This paper aimed to conduct a systematic review to synthesize the findings regarding eating behavior changes during the COVID-19 pandemic. The topic is important. However, several issues need to be addressed:
Introduction. Different countries implemented different measures to tackle the challenges during the COVID-19 pandemic. For example, school closures due to the pandemic have impacted the school meal programs which aim to provided nutrition support for children from economically disadvantaged households. New ways of distributing such nutrition assistance programs benefits were implemented. Dietary behaviors and the quality of certain populations have been disproportionately impacted due to these changes. 1-2 paragraphs are needed in the introduction section to describe these changes to better justify the significance and motivation of the paper.
Methods. It is unclear what time frame the authors used for the searches. The authors mentioned, “The search was first piloted in the PubMed database on October 7, 2020 (as shown 73 in Table 2) and filtered since 2019.” What date in 2019? Also, the searches should cover 2021 as well. I suggest the authors expanding the searches to the most recent publications, at least until June 30, 2021.
Results. Table 3: Please add information about where the study was conducted (country, city/county, town, etc). Also, under design type, we see “Observational cohort”, “Cohort”, “Longitudinal cohort”, “observational”, etc. What are the differences between these labels? We know cohort studies are in general longitudinal observational studies, either prospective or retrospective. Please provide clear definitions of these. Be careful and clear when using epidemiological terms.
Table 4: The Results column needs to be edited and condensed. Too wordy. Needs to be condensed.
Lines 188-190: “To quantify the degree of agreement, intraclass correlation (ICC) analysis and the 188 Kappa Index were calculated. The results showed that there was a strong agreement level 189 (ICC= 0,88, p= 0.00; Kappa=0,875, p= 0.00).”. It is unclear how the ICC was estimated, eg ICC for an overall outcome, or for a specific outcome such as processed food daily intake? More detailed information is needed to help readers interpret the results.
Discussion. The current discussion is too thin. The authors should elaborate more about the possible mechanisms underlying the findings, and more importantly, policy implications.
Round 2
Reviewer 1 Report
significant changes have been made according to the reviewers' comments and statements are more appropriate considering multiple factors influencing eating behaviors.
Reviewer 2 Report
First of all, I would like to respect the efforts of the authors in revising the paper. I think the authors have revised the paper well and it is now worthy of publication.
Reviewer 3 Report
Agree.